# Effect of Thermal Treatment and Erosion Aggressiveness on Resistance of S235JR Steel to Cavitation and Slurry

**DOI:** 10.3390/ma14061456

**Published:** 2021-03-16

**Authors:** Alicja K. Krella, Dominika E. Zakrzewska, Marta H. Buszko, Artur Marchewicz

**Affiliations:** Institute of Fluid Flow Machinery PAS, Fiszera 14, 80-231 Gdańsk, Poland; dzakrzewska@imp.gda.pl (D.E.Z.); mbuszko@imp.gda.pl (M.H.B.); artur.marchewicza@imp.gda.pl (A.M.)

**Keywords:** cavitation erosion, slurry erosion, carbon steel, hardness, surface roughness, fracture, resistance

## Abstract

S235JR steel is used in many applications, but its resistance to the erosion processes has been poorly studied. To investigate this resistance, cavitation, and slurry erosion tests were conducted. These tests were carried out at different erosion intensities, i.e., different flow rates in the cavitation tunnel with a system of barricades and different rotational speeds in the slurry pot. The steel was tested as-received and after thermal treatment at 930 °C, which lowered the hardness of the steel. To better understand the degradation processes, in addition to mass loss measurements, surface roughness and hardness were measured. Along with increasing erosion intensity, the mass loss increased as well. However, the nature of the increase in mass loss, as well as the effect of steel hardness on this mass loss, was different for each of the erosion processes. In the cavitation erosion tests, the mass loss increased linearly with the increase in flow velocity, while in the slurry tests this relationship was polynomial, indicating a strong increase in mass losses with an increase in rotational speed. Cavitation erosion resulted in stronger and deeper strain hardening than slurry. Surface damage from cavitation erosion tests was mainly deep pits, voids, and cracks during the slurry tests, while flaking was the most significant damage.

## 1. Introduction

Hydraulic devices, as well as other devices operating in a liquid, e.g., ship propellers, turbines, pumps and pipelines, are exposed to cavitation erosion, solid particle erosion, corrosion, as well as to the combined effect of these types of destruction. Cavitation erosion and solid particle erosion are the degradation processes that play a significant role in the service life of said devices. Damage of the devices results in a loss of their performance, as well as high repair costs and downtime costs. The mentioned degradation processes usually act separately, or one process is dominating. Although in both destruction processes, the degradation results from repeated impacts, the mass loss and the erosion rate can be substantially different. However, there is very little research showing differences in the destruction process. For this reason, it is desirable to know the development of material degradation and the resistance of the material to each of the degradation processes mentioned.

Erosion resistance tests are conducted mainly for corrosion-resistance materials, such as stainless steels or titanium alloys due their application in many hydraulic devices. Investigations of titanium alloys (CP-Ti with a hardness of 219 HV, Ti5111 with a hardness of 265 HV, Ti-6Al-4V ELI with a hardness of 342 HV and Ti-6Al-4V ELI/Ru with a hardness of 348 HV) showed that the alloy with the best cavitation erosion resistance does not have the best slurry erosion resistance [1]. The best erosion resistance in slurry tests had Ti5111 alloy, while in cavitation erosion tests the best resistance had Ti-6Al-4V ELI/Ru alloy [1]. Similar results were obtained by Schillo et al. [2]. Investigations of Co-based coating and Ni-based coating showed better resistance of the Co-based coating to cavitation erosion, while the Ni-based coating to slurry erosion [2].

Because of high costs of stainless steels, carbon steels are still used for some ship elements and hydraulic devices, despite their low corrosion resistance [3,4]. They are used because of good weldability, good strength properties and low price [5,6]. Nevertheless, carbon steels have an unsatisfactory amount of research into these types of degradation. Carbon steels have been the subject of investigations by Hattori et al. who developed a database of their cavitation erosion resistance [7]. Also, Tzanakis et al. [8] have investigated the cavitation erosion resistance of carbon steels. They have obtained that the steel with the lowest hardness, yield and tensile strength had the lowest cavitation erosion resistance. However, steel with the highest hardness and tensile properties did not had the best resistance. Moreover, the test duration highly affected which steel had the best cavitation erosion resistance (the lowest mass loss). In the case of slurry erosion, carbon steels, due to their use for pipelines, have been the subject of investigations by Alam and Farhat [9], Islam et at. [10], Wood et al. [11] and also by Ojala et al. [12]. An investigation of API X42, API X70, and API X100 steels showed that, with increasing hardness and yield strength, the slurry resistance increased as well [10]. Similar results were obtained by Ojala et al. [12]. However, later investigations of a wider range of carbon steels (AISI 1018, AISI 1080, API X80, X100 and API X120 steels) did not confirm such relations [9]. Another common steel used in many industrial applications is S235 JR steel due to its very good weldability, good mechanical properties and low price [5,6]. Ultrasonic impact treatment applied with various duration (5, 15, and 30 min) on this steel increased hardness from 133 HV to 213, 214 and 224 HV (by 60.1%, 60.9%, and 68.4%) at top of the surface layer, respectively, and decreased mass loss by about 15%, 27%, and 50%, respectively, in wear tests [6]. Thus, a slight difference in steel hardness (from 213 HV to 224 HV) caused an essential increase in wear resistance. Despite many investigations performed, there is a lack of investigation of carbon steel for resistance to both cavitation erosion and slurry erosion. For this reason, such investigations have been performed.

As mentioned, most investigations of materials pointed to hardness as the property that plays an essential role in resistance to cavitation erosion and slurry erosion. An increase in hardness causes an increase in erosion resistance (decrease in mass/volume loss in erosion tests). However, the effect of steel hardness on the resistance to cavitation erosion or slurry erosion is related to the test conditions, i.e., intensity of dynamic loading. In the case of cavitation erosion testing, the intensity of dynamic loading depends on flow velocity and the shape of cavitation inducers for flowing cavitation and on frequency and amplitude of vibration in ultrasonic cavitation [13,14,15]. While in the case of slurry erosion tests, the intensity of dynamic loading depends on the flow velocity as well as the size, shape and concentration of solid particles [16]. Many factors affecting the intensity of dynamic loading, which in turn affect resistance to cavitation and slurry erosion, indicate the need for experimental investigation.

The aim of the study was to investigate the effect of the type of erosion (cavitation and slurry) on the strength and degradation mechanism of S235JR carbon steel, and to show the differences in the destruction process between these types of erosion. S 235JR steel was chosen for its strong influence of hardness on mass loss and erosion resistance, as demonstrated by Kahraman et al. [6]. For this reason, the S235JR carbon steel was tested in two states: as received and after heat treatment, which was applied to obtain different hardness of the steel. Studies were carried out at various erosion intensities to analyze their effect on cavitation and slurry erosion rate.

## 2. Materials and Methods

The chemical composition of S235JR carbon steel obtained from the steel certificate is shown in Table 1. S235JR steel is low-carbon ferritic-perlitic steel with little content of Mn. S235JR steel was investigated in two states: as-received and after thermal treatment. The most popular thermal treatment is annealing. The temperature of annealing (930 °C) was chosen based on investigations performed by Rdzawski et al. [17]. Thermal treatment was performed using Nabertherm L3/11 furnace (Nabertherm, Lilienthal, Germany). The steel specimens were marked as S235JR AR and S235JR TT, respectively, for the specimens in the as-received state and the specimens after thermal treatment.

In order to investigate the effect of cavitation intensity on the erosion process, four tests were carried out using a cavitation tunnel equipped with a system of barricades. The scheme of the cavitation tunnel is shown in Figure 1 and it has been described by Krella et al. [18]. As a working fluid tap water with a temperature of 20 ± 2 °C was used. The intensity of cavitation is governed by the fluid flow rate, which is controlled by the inlet and outlet pressures, which in turn are regulated by the inlet and outlet valves. Flow speed was measured using a portable ultrasonic flowmeter Fluxus^®^ model ADM 6725 (FLEXIM, Edgewood, NY, USA) with a measurement accuracy of ±0.5% of the measured value. In the presented investigations, the water flow velocities were 35 m/s, 37.9 m/s, 40.5 m/s, and 43 m/s in the slot between the barricades. The total duration of the test was 600 min.

In order to obtain the cavitation curves and erosion rates, the mass of tested specimens was measured using an analytical balance of 0.1 mg accuracy. Mass measurements were carried out before the tests and after each exposure. Before each mass measurements, the specimens were cleaned and dried. The shape and dimensions of the specimens are shown in Figure 2.

The slurry tests were performed using a slurry pot tester (Figure 3). The pot capacity is 6.4 l. The tester allows simultaneous examination of two specimens. At the inner wall of a slurry pot, four baffles are mounted to minimize centrifuging of the fluid due to the rotation of the specimens. Additionally, a stirrer is mounted to prevent the sedimentation of the solid particles. The rotation speed is adjusted by a power inverter ABB ACS 300 (ABB Asea Brown Boveri Ltd., Zurich, Switzerland). To analyze an effect of erosion intensity on the erosion process, four slurry tests were carried out at rotational speeds of 565 rpm, 786 rpm and 1012 rpm. The duration of the slurry tests was 600 min, similar to cavitation erosion tests. Tap water with a temperature of 18 °C was used as a working liquid. Spherical steel solid particles with a diameter of 520 µm and hardness of 51 HRC were used as erodents. The concentration of erodent was 12.5 wt.%. The impact angle was 90°. The shape and dimensions of the specimens are shown in Figure 4. Before the tests and after each exposure, the specimens were cleaned, dried, and reweighted using an analytical balance of 0.1 mg accuracy.

After each erosion test, the microscopic studies were carried out using a scanning electron microscope EVO-40/Zeiss (Oberkochen, Baden-Württemberg, Germany) and a SU3500 scanning electron microscope, Hitachi (Tokyo, Japan), in the Research Centre in Jabłonna, Poland. Moreover, the microhardness measurements on the cross-sections were carried out using the FALCON 401 Vickers Hardness (INNOVATEST, Shanghai, China) tester with 300 gf load and dwell time of 10 s. The first measurement was made 35 µm from the eroded surface and the measurements were carried out at the distance of 200 µm to eliminate an influence of the other indents on the result obtained. Moreover, to characterize the eroded surfaces, surface roughness (Ra parameter) was determined using the SJ-301 Mitutoyo Surface Roughness Tester (Mitutoyo Corporation, Kanagawa, Kawasaki-shi, Japan). The Ra parameter is an arithmetical mean of the absolute values of the profile deviations from the mean line of the roughness profile. In the case of the cavitation erosion tests, the surface roughness was measured after the tests along the scheme of the network shown in Figure 5. In the case of slurry erosion tests, the specimen surface was eroded evenly. Therefore, the surface roughness was measured at five arbitrary chosen places. Additionally, surface roughness measurements were performed after each test exposure.

## 3. Results

### 3.1. Thermal Treatment

Thermal treatment affects the hardness and structure of S235JR steel. The performed measurements of steel hardness have shown that thermal treatment decreased steel hardness (surface hardness). The initial surface hardness was 178.7 ± 3.5 HV of the specimens used for cavitation erosion test and 151.74 ± 0.86 HV of the specimens used for slurry erosion tests. The reasons for the different initial hardness values were different steel sheets from which the test specimens were made. After thermal treatment, the surface hardness was 144.5 ± 2.9 HV and 109.3 ± 1.13 HV, respectively. Thus, surface hardness decreased 34 HV and 42.5 HV, respectively. This shows that a decrease in surface hardness is comparable. Any changes in surface hardness caused during erosion tests should also be comparable.

### 3.2. Cavitation Erosion Test

The results of cavitation erosion tests are shown in the form of the cavitation erosion curves (Figure 6) and the cavitation erosion rate curves (Figure 7).

These curves show that a decrease in steel hardness and an increase in flow velocity increased the mass losses and erosion rates. These results are consistent with the expectation and investigation obtained by Kahraman et al. [6]. The decrease in steel hardness by 34.2 HV as a result of thermal treatment caused an increase in mass loss after the entire tests by about 43 mg to about 49.7 mg, depending on the test conditions. Thus, the high impact of the hardness of this steel on the erosion resistance was confirmed, although the tests carried out were tests of cavitation erosion, not slurry. However, an increase in flow velocity did not cause a proportional increase in mass loss of the tested steel, either in the as-received state or after thermal treatment (Figure 6). Moreover, the shape of the cavitation erosion curves indicates the different degradation processes. In the case of the S235JR-AR steel, the cavitation erosion curves indicate two main erosion periods: the first one lasting about 300 min and the second starting from about 360 min. While in the case of the S235JR-TT steel after thermal treatment, the mass losses increased more or less evenly throughout the cavitation erosion tests. This also confirms a high effect of the hardness of this steel on the degradation process.

Figure 7 shows that the mass loss of the tested steel started with a high erosion rate, regardless of the steel hardness. Only the thermal-treated steel tested at 35 m/s failed this rule. After reaching the maximum value, the erosion rate decreased. In the case of the S235JR-AR steel, the erosion rate decreased only until about 180 min of testing. After that, the erosion rate increased. In the case of S235JR-TT steel, the most rapid decrease in erosion rate occurred during the initial 30 min and then the erosion rate decreased more slowly. The period of erosion rate decrease depended on the flow velocity. In the case of the tests carried out at 35 m/s and 37.9 m/s, this period lasted until about 300 min and then the erosion rate slightly increased. In the case of the tests carried out at 40.5 m/s, the period of the erosion rate decrease lasted until about 360 min. While in the case of the tests carried out at 43 m/s, this period lasted until the end of the test, but after 360 min of testing, this decrease was much slower. Figure 7 also shows that the greatest difference in erosion rate was at the start of the study to approximately 180 min, and as the duration of the study increased, the difference in erosion rate decreased. Thus, the influence of steel hardness on the erosion rate decreased with the duration of the tests. This indicates changes in the structure and properties of the tested steel, i.e., hardness, as a result of impacting the micro-jets formed during the implosion of cavitation bubbles.

### 3.3. Slurry Erosion Test

The results of slurry erosion tests are shown in the form of the erosion and erosion rate curves (Figure 8 and Figure 9, respectively). The maximum mass losses obtained in the tests carried out with the highest rotational speed of 1012 rpm (Figure 8) are slightly higher than those obtained in cavitation erosion tests, i.e., tests performed at a flow velocity of 43 m/s (Figure 6). This indicates that the maximum rotational speed was chosen correctly. The mass losses increased along with the increase in rotational speed. The increase from 565 rpm to 786 rpm resulted in more than a 12-fold increase in the mass loss of the as-received steel, while a further increase in speed to 1012 rpm only resulted in an almost 3-fold increase. In the case of thermal-treated steel, the increase in rotational speed caused a 6.7-fold and 2.5-fold increase in mass loss, respectively. Thus, the influence of steel hardness on the mass losses is visible. Nevertheless, the slurry erosion curves have a completely different shape than the cavitation erosion curves. Moreover, in the initial periods of tests, a decrease in mass loss was noted for steel after thermal treatment with lower hardness compared to steel in the as-received state with higher hardness. However, after some time of testing, the mass losses of S235JR-TT steel became greater. The duration of the test, in which the mass loss of the S235JR-TT steel became greater than that of the S235JR-AR steel, increased with increasing rotational speed, which was unexpected. The obtained results, especially a decrease in mass loss of the steel with lower hardness, were in opposition to those obtained in cavitation erosion tests and by Kahraman et al. [6], who also tested the resistance of this steel to slurry erosion.

In the case of the tests performed at 565 rpm and 786 rpm, the mass loss difference in the initial period was very slightly, in fact, it was in the measurement error. The increase in mass loss of the S235JR-TT steel in relation to the S235JR-AR steel occurred after 120 min and 160 min, respectively, and then this difference increased. For tests performed at the highest rotational speed (1012 rpm), this difference in mass losses was already significant. The highest (10 mg) occurred between 240 and 300 min of testing. The time when the mass loss of the S235JR-TT steel become bigger than that of the S235JR-AR steel was about 550 min. Thus, the difference in mass losses of S235JR-TT steel and S235JR-AR steel, and the test duration, when the mass loss of the S235JR-TT steel become bigger than for the S235JR-AR, increased with increasing rotational speed.

The influence of thermal treatment on the degradation process is more visible when analysing erosion rates (Figure 9). In the case of tests performed at 565 rpm and 786 rpm, the erosion rate of the S235JR-AR steel increased to the maximum value that remained unchanged until the end of the tests. In the case of the S235JR-TT steel, such a stable erosion rate was not observed. Thus, the lower hardness of the S235JR-TT steel and changes in the structure caused by the action of erodents made it impossible for this steel to achieve a stable erosion rate. The erosion rates increased through the entire tests, but the intensity of this increase was related to rotational speed.

In the case of tests performed at 1012 rpm, the erosion rate of the S235JR-AR steel increased to the maximum value, which was obtained after 90 min of testing, and then decreased. The process of erosion rate decrease lasted until the end of the test. The decrease in erosion rate was likely caused by a surface layer hardness increase. In the case of S235JR-TT steel, the erosion rate increased rapidly during the first 60 min of testing, then it slightly decreased. After 240 min, the erosion rate increased again, but after about 480 min, this increase accelerated. This shape of the erosion rate curve indicates that the surface strain hardening process did not occur or was very small, despite the low hardness of this steel. The high intensity of the dynamic loading probably caused a much faster crack growth than the strain hardening of this mild steel. Thus, the steel hardness and the intensity of loading strongly influenced the degradation mechanism.

### 3.4. Hardness

In the case of cavitation and slurry erosion, degradation occurs as a result of repeated impacts. The erosion curves (Figure 7 and Figure 9) show changes in the properties of the steel, in particular the surface hardness. For this reason, a detailed study of the influence of the steel state and test conditions on changes in surface hardness is important for the improvement of knowledge about erosion processes. The changes in hardness caused by cavitation and slurry at the highest and lowest erosion intensity along the specimen depth are shown in Figure 10 and Figure 11, respectively. As the initial hardness of the specimens used for cavitation tests differed from the specimens used for the slurry tests, the strain hardening effect was also determined and shown in Figure 10 and Figure 11. The strain hardening effect after erosion tests was calculated from the following formula:(1)$$\Delta HV=\frac{HV_{x}-HV_{0}}{HV_{0}}\cdot 100\%$$
where *HV_x_* is the hardness at the distance of x and *HV_0_* is the initial hardness.

Although mass losses of S235JR-AR steel tested at the highest erosion intensity were comparable, the surface hardness changes and the hardening effect were different (Figure 10 and Figure 11). The effect of cavitation on surface hardness was more pronounced than that of the slurry. This is evidence of the different degradation mechanisms involved in any erosion process.

In the case of the cavitation erosion tests conducted at the highest flow velocity of 43 m/s, the hardness of the S235JR-AR steel increased up to 194 HV at a depth of 35 μm, but the maximum value was 197 HV at a depth of 235 μm (Figure 10a). Such a hardness peak underneath the surface was also obtained in the tests of Fe–Cr–Ni–Mn stainless steels [19,20]. This makes that the maximum strain hardening effect determined from Equation (1) was 10% at a depth of 235 μm (Figure 10b). Moreover, the hardening effect was noted down to a depth of 4 mm. In the case of the S235JR-TT steel tested for resistance to cavitation erosion in these conditions, the maximum increase in hardness was only about 8% (157 HV) at a depth of 235 μm and the hardening effect was noted down to a depth of 2.5 mm. Thus, the initial hardness of the steel prior to the cavitation test had an effect on the hardenability. Steel with lower initial hardness underwent less hardening, and steel with higher initial hardness—stronger. Moreover, as the strengthening of the surface layer increased, the mass loss (Figure 6) and the erosion rate (Figure 7) decreased.

In the case of the tests carried out at the lowest cavitation erosion intensity, i.e., at a flow velocity of 35 m/s, the hardness changes of the S235JR-AR steel were very slight. The hardening effect was in the range from −0.5% to 1%. This shows that no hardening effect occurred. However, tests of the S235JR-TT steel showed a strain hardening of 5% at a depth of 35 μm. With increasing depth, hardening ranged from 2% to 5%. The total depth of hardening was 4 mm. Thus, at the flow velocity of 35 m/s, the supplied impact energy resulted in the opposite development of the degradation process to that at the flow velocity of 43 m/s. This is due to a change in the distribution of the cavitation load—the number of high-amplitude pulses and the number of low-amplitude pulses. In the case of harder S235JR-AR steel, only the initiation and growth of cracks leading to mass loss was noted. It was caused by high-amplitude pulses, low-amplitude pulses were not able to cause any effect because of high hardness of the tested steel. Meanwhile, in the S235JR-TT steel with a lower hardness, strain hardening was also noted despite greater mass loss (Figure 6). Strain hardening was likely an effect of large number of low-amplitude pulses and low hardness of the tested steel.

In the case of the slurry erosion tests performed at the highest rotational speed of 1012 rpm, the hardness of the as-received steel and the thermally treated steel increased to about 159 HV and 120 HV at a depth of 35 μm (Figure 11a). Thus, the strain hardening effect was about 5% and 8%, respectively (Figure 11b). This indicates that a higher strain hardening effect was obtained for the thermally treated steel, which had lower initial hardness. This effect was in opposition to that obtained during cavitation erosion tests at the flow velocity of 43 m/s. Moreover, the strain hardening of the as-receives steel was much lower than that obtained in cavitation erosion tests, while the thermally treated steel was comparable to that obtained in cavitation erosion tests (Figure 10b). The depth of strain hardening was about 1800 µm and 1400 µm, respectively. However, the biggest decrease in strain hardening occurred at a distance of about 200 µm (from 8.2% at 35 µm to 0.7% at 235 µm). These depths are much smaller than those obtained in cavitation tests (about 4 mm). This indicates the different degradation mechanism. In cavitation erosion, cavitation pulses impact the surface with a wide range of amplitudes, from a few kPa to GPa. In slurry erosion, all erodents impact with comparable energy, so the dispersion of impact energy is small. Due to the high rotational speed, which influenced the high impact velocity, the impact energy is used mainly on the initiation and development of cracks, resulting in a significant mass loss and less strain hardening than in the case of cavitation erosion, where low energy impacts also occurred.

In the case of tests carried out at the lowest slurry erosion intensity, i.e., at a rotational speed of 565 rpm, the hardness of the S235JR-AR steel at a depth of 35 µm was about 152 HV. It means that the hardness increase did not occur. All changes in hardness oscillated around the initial value. In the case of S235JR-TT steel, the impacts of solid particles caused a slight increase in hardness to about 112 HV at a depth of 35 µm. The strain hardening effect was about 1%. The highest hardness (113.4 HV) was obtained at a depth of 635 µm. This makes that strain hardening increased to 2.2%. Then, hardness decreased to an initial value that was reached at a depth of about 1800 µm. Thus, the depth of strain hardening was bigger than in the tests performed at the highest rotational speed. The likely reason for this was the lower impact energy supplied by erodents to the steel surface that caused significant less mass loss and much less removal of the hardened layer.

### 3.5. Surface Roughness

The distributions of surface roughness along the network shown in Figure 5 that developed during the entire cavitation erosion tests are shown in Figure 12 and Figure 13, respectively for S235JR-AR and S235JR-TT steel. In order to show the effect of flow conditions on the roughness, the same maximum Ra value was used in the color bar despite the different maximum Ra values. During the slurry test, the entire surface of the specimens was uniformly eroded, so five measurements of the Ra parameter were taken after each exposure. The development of the Ra surface roughness parameter is shown in Figure 14.

Roughness measurements of specimens tested in cavitation erosion tests revealed areas of high roughness Ra > 5 μm that are at approximately half the width of the specimens close to the upper barricade (Figure 12 and Figure 13), regardless of the hardness of the tested steel (steel state) and the flow velocity. With increasing flow velocity, the area with Ra > 1 μm also increases, but it is not followed by the increase in the maximum roughness value. Also, the distribution of Ra parameters changed its character depending on the flow velocity. In the case of tests conducted at a flow velocity of 35 m/s, the maximum Ra value was 7.38 ± 0.16 μm and 8.71 ± 0.21 μm, respectively for S235JR-AR and S235JR-TT steel. Thus, higher roughness was obtained on softer steel. The surface roughness increased rapidly to a maximum value, which was reached 5 mm from the specimen edge nearest to the upper barricade, and then decreased, regardless of the steel state. However, the rate of decrease in roughness was related to the hardness of the steel. In the case of softer steel after thermal treatment, this rate was lower. Although the maximum surface roughness had higher values, the roughness (Ra parameter) of less than 1 μm was only recorded at a distance of 20 mm from the upper barricade. Nevertheless, in both steel states, the roughness distribution formed a sharp peak.

For the tests carried out at a flow velocity of 37.9 m/s, the maximum Ra value was 8.47 ± 0.15 μm and 11.60 ± 0.27 μm, respectively (Figure 12b and Figure 13b). Thus, the maximum Ra value increased 1 μm and nearly 3 μm, respectively. Moreover, the area with Ra > 2 μm increased significantly. The roughness peak was still obtained and located 5 mm from the upper barricade. The roughness peak was sharp in S235JR-TT steel (Figure 13b), while slight in S235JR-AR steel (Figure 12b). Moreover, the eroded surface with Ra > 6 μm formed a large area indicating a zone of high erosion level.

A further increase in a flow velocity to 40.5 m/s caused an increase in the area with Ra > 2 μm. However, the roughness values lowered concerning those obtained at a flow velocity of 37.9 m/s. The maximum Ra values of S235JR-AR and S235JR-TT steel were 7.71 ± 0.19 μm and 9.38 ± 0.28 μm, respectively. Thus, the maximum Ra value decreased by nearly 0.8 μm and over 2 μm, respectively. Considering that with increasing flow velocity the mass loss also increased (Figure 6), a decrease in surface roughness was caused likely by overlapping damages. However, the surface with Ra > 6 μm increased without any roughness peaks indicating a comparable erosion level.

In the case of tests carried out at a flow velocity of 43 m/s, surface roughness decreased even more (Figure 12d and Figure 13d). The maximum Ra values were 4.31 ± 0.11 μm and 5.48 ± 0.12 μm, respectively. Thus, the Ra parameter decreased by nearly 3.5 μm and 4 μm, respectively. Besides, the Ra parameters were lower than those obtained in the tests carried out at the lowest flow velocity. This decrease was caused by the effect of overlapping damages. The surface with the highest surface roughness formed a large, flattened area with a comparable erosion level. The conducted tests showed that the steel of lower hardness was characterized by higher roughness regardless of the flow velocity. This is in agreement with the mass losses and erosion rates.

Slurry erosion tests also caused an increase in surface roughness (Figure 14). The highest increase took place during the first 60 min of testing. After reaching the maximum value, the Ra parameter remained stable until the end of the tests. The obtained fluctuations result from the nature of the roughness measurement and should be disregarded in further analysis. The S235JR-TT steel with lower hardness had higher Ra parameters, regardless of the test conditions. This result is comparable to the result obtained in cavitation erosion. Nevertheless, the highest Ra parameters were much lower than those obtained in cavitation erosion tests. In the case of the tests carried out at a rotational speed of 565 rpm, the Ra parameters of S235JR-AR steel and S235JR-TT steel after the entire tests were 0.29 ± 0.04 μm and 0.33 ± 0.06 μm, respectively. In the case of the tests performed at a rotational speed of 786 rpm and 1012 rpm, the Ra parameters were 0.36 ± 0.03 μm and 0.42 ± 0.03 μm, and 0.44 ± 0.04 μm and 0.54 ± 0.03 μm, respectively. Thus, the Ra parameter increased with the increase of the rotational speed, as well as the difference in the surface roughness values. Thus, there was no overlapping damage effect, as was the case with cavitation erosion.

### 3.6. Microscopic Observation

Based on the roughness measurements, the zones of low and high erosion in cavitation tests were determined and showed in Figure 15. The surface damages of the S235JR-AR and S235JR-TT steels developed during tests performed at the lowest (35 m/s) and highest (43 m/s) flow velocity in these zones are shown in Figure 16 and Figure 17, respectively.

Damages to the surface of hard S235JR-AR and soft S235JR-TT steels in the strongly eroded zone and tested at a flow velocity of 35 m/s have the form of pits or so-called cavitation tunnels, voids and cracks developing in a ductile mode (Figure 16a,b). Cracks are a bit wider for soft S235JR-TT steel, indicating the effect of the steel hardness. Although the diameter of some voids or pits can be determined, all damages overlapped. Besides, the damage mentioned above is greater for soft S235JR-TT steel, which complies with the roughness measurements (Figure 12 and Figure 13) and steel hardness. Ductile fracture is in line with the steel grade (low carbon steel) and low hardness (178.7 ± 3.5 HV and 144.5 ± 2.9 HV, respectively) and ferritic-pearlitic structure. In the case of the low erosion zone (Figure 16c,d), the damage is mainly in the form of pitting, but several short cracks are also observed. The damage density is much lower on the surface of the S235JR-AR steel, which had a higher hardness. The pits are well seen and can be divided into two groups: the first with a diameter up to 10 μm and the second with a bigger diameter (above 10 μm). The pits of the first group are much deeper than those of the second. This indicates that the pitting of the first group was the result of shearing by micro-jets impacted at high speed. Such pits do not occur on the surface of hard S235JR-AR steel. Considering that the tests were carried out at the same condition, the S235JR-AR was more resistant to such impacts because of steel hardness.

Damages on the surface of S235JR-AR and S235JR-TT steel in the strongly eroded zone and tested at a flow velocity of 43 m/s also have the form of ductile cracks, cavitation pits/tunnels and voids (Figure 17a,b). Thus, all these failures are typical of this steel and this erosion process. Nevertheless, the intensity of cavitation affected the number and width of the cracks, which are much narrower than those observed at a flow rate of 35 m/s. The diameter of some pits/cavitation tunnels in hard S235JR-AR steel ranges from about 10 μm to 20 μm (Figure 17a), but such pits are not seen in soft S235JR-TT steel (Figure 17b). It is related to the hardness of the steel and the mass losses. Taking into account the greater loss of mass on the S235JR-TT steel in the cavitation test, the most likely reason for the lack of such pitting is the removal of such a heavily damaged surface layer. On the other hand, the damage size of the steel as received is smaller than that of thermal-treated steel, which is consistent with the hardness measurements (Figure 10). Also, more severe damage developed during tests with a flow velocity of 35 m/s is consistent with the hardness measurements. Moreover, the observation of the degree of destruction of S235JR steel in both states is consistent with the roughness measurements (Figure 12 and Figure 13).

In the case of the low erosion zone (Figure 17c,d), the damage is mainly in the form of pitting, but several cracks and surface undulation are also observed. The cracks are likely developed along the grain boundaries. The length and width of cracks, as well as the size of pits, are much bigger for the thermal-treated steel. The pits were caused by shearing due to the impacts of high-velocity micro-jets. The bigger size of pits indicates that this steel is more susceptible to such impacts. However, damage density is much bigger for steel in the as-received state. The lower size of pits and cracks length are connected with a higher hardness of this steel.

The damage from slurry erosion testing is completely different (Figure 18). In the case of tests carried out at a rotational speed of 1012 rpm (Figure 18a,b), flaking and undulation of the surface are visible, regardless of the steel state. There are also some tiny and shallow pits. The surface is much less degraded concerning that developed in cavitation tests (Figure 16 and Figure 17). The level of surface degradation is consistent with the roughness results (Figure 14). Also, damage developed at a rotational speed of 565 rpm degradation is consistent with the roughness results (Figure 14). As shown in Figure 18c,d, the surface of S235JR-AR steel is slightly undulated, similar to the surface of the thermal-treated steel. However, the undulation of the latter (Figure 18d) seems to be bigger. Yet, the surface looks much smoother. There is no flaking, but pits and short cracks. The number and size of pits depend on the steel state. In the case of the as-received state, pits are much smaller, but their number is much bigger than on the surface of S235JR-TT steel (Figure 18c,d). The pit seen in Figure 18d was likely due to steel removal and not shearing as with cavitation erosion.

## 4. Discussion

The performed thermal treatment of the tested steel decreased the hardness by over 30 HV and allowed learning the impact of steel hardness on the resistance of S235JR steel to cavitation and slurry erosion. According to the test results obtained by Hammit [21], Kwok et al. [22] and Rajput et al. [23], resistance to cavitation erosion increases exponentially with increasing hardness. Similar correlation occurs for slurry. Sheldon [24] and Desale et al. [25,26] obtained that resistance to slurry erosion increases exponentially with increasing hardness. Based on the data obtained by Kwok et al. [22] and Rajput et al. [23], this exponent is close to 3. However, when this exponent was determined on the basis of the results obtained by Baghel et al. [27], it was close to 1, while Moore’a [28]—to 0.4. Nevertheless, the exponent can range from about 0.5 to over 3 in both types of erosion. On the other hand, investigations of titanium alloys have not confirmed this relationship either in cavitation erosion or in slurry [1], as well as tests of X10CrAlSi18 steel annealed at different temperatures [29]. Slurry tests of aluminum, copper, carbon steels and stainless steels confirmed that material hardness and impact velocity affect the erosion rate [30]. However, no clear single trend of change was achieved. Depending on the tested materials with a given hardness range, the erosion rate may decrease with increasing hardness, but it may also increase, or no effect may be obtained. Kahraman et al. [6], who tested the same steel, have obtained a significant decrease in mass loss in the slurry tests with the increase in steel hardness. The results obtained in these investigations did not confirm such a relationship in the study of slurry erosion (Figure 8a), but showed a big impact of steel hardness on the mass loss in cavitation erosion tests (Figure 6). To better see the effect of steel hardness on the mass loss, a graph was made which takes into account the mass loss obtained at the highest flow velocity (43 m/s) or rotational speed (1012 rpm) after 600 min (the entire tests) and e.g., after 300 min, when the highest difference in mass losses occurred. Figure 19 shows that for S235JR steel, steel hardness strongly influenced the cavitation resistance, i.e., mass loss. The hardness change by 34 HV caused a change in the mass loss by about 40 mg (38 mg after 300 min of testing and 43 mg after the entire tests). Thus, this relationship was slightly dependent on the duration of the test. In the case of slurry tests, the decrease in hardness from 151.7 HV to 109 HV slightly affected mass loss and erosion resistance. Similarly, such a result was independent of the duration of the test. This result was unexpected in the light of the tests carried out by Kahraman et al. [6] and also by Desale et al. [25,26]. However, Oka and Yoshida [30] found that for some materials, some hardness values and specific types of erodent, there may be no change in mass loss caused by a change in the hardness of the steel. Thus, the obtained difference in hardness and mass losses shown in Figure 19 fit in with that observation. Figure 19 shows also that the steel hardness plays a much stronger influence on the mass loss in cavitation erosion resistance tests than in the case of slurry erosion.

The other factor influencing the mass loss in cavitation and slurry erosion tests is erosion aggressiveness/intensity. In the case of cavitation erosion, the intensity of the erosion depends on the flow velocity or inlet and outlet pressures in the flow devices, e.g., in a cavitation tunnel or a cavitating jet device, and the frequency and/or amplitude of the vibrations in the vibrating device. Investigations of pure aluminum, copper, bronze, carbon steels and also stainless steels with the use of the cavitating jet apparatus showed an exponential dependence the mass losses with injection and tank pressures [31]. While, in the case of the vibrating apparatus, the relationship between vibrating amplitude and mass losses of pure aluminum and cooper was linear [15,32]. In the case of slurry erosion, the impact velocity of erodents is the parameter that affects the erosion intensity and the mass loss the most [26,33,34,35]. Moreover, erosion rate increases exponentially with the impact velocity and an exponent is usually about 2–3 for metallic alloys, but it can also vary from 0.34 to 4.83 [35]. The correlations between the flow velocity in cavitation erosion tests or rotational speed in slurry tests and the mass loss after the entire erosion tests obtained in these investigations are shown in Figure 20. In the case of cavitation erosion tests, a linear relationship for both steel states was obtained for the tests carried out in the cavitation tunnel and flow velocities ranging from 35 m/s to 43 m/s (Figure 20a). This linear correlation is in accordance with the results obtained by Okada et al. [15] and Hattori et al. [32], although other testing materials and various apparatus were used. However, based on the results obtained by Krella [36], it can be seen that the type of correlation depends on the flow velocity range, and the linear correlation may occur above a certain threshold value of the flow velocity. Figure 20a also shows that the trend line determined for hard S235JR-AR steel is parallel to that determined for the soft steel after thermal treatment. This means that the difference in the mass losses obtained after the entire tests was comparable for all flow velocities and the character of the trend line was independent of the steel state and steel hardness.

In the case of slurry tests, for rotation speed range from 565 rpm to 1012 rpm, the best relationship is a polynomial relationship, not exponential (Figure 20b). In the case of the exponential dependence, the maximum dispersion may reach even 42% for steel in the as-received state or 17% for heat-treated steel. This indicates that the exponential relationship cannot be taken into account. The trend lines of hard S235JR-AR and soft S235JR-TT steels are parallel to each other, so the nature of the correlation does not depend on the state of the steel and steel hardness, as was the case with cavitation erosion, but on the steel grade. The other factors that may affect the relationship are the type of test device, erodent concentration and the range of rotational speed. Also, in the tests of ferritic X10CrAlSi18 steel with the use of a slurry pot, no exponential but rather a linear correlation was obtained by Krella et al. [37]. In a slurry pot, the solid particles can collide with each other, which affects the actual speed and direction of the particles. As speed increases, this interaction is likely to increase. A low depth of strain hardening at the highest rotational speeds (Figure 11) is consistent with the assumption that the particles interact with each other.

Although erosion processes concern mainly the degradation of the material surface layer, the investigations of surface roughness development in relation to erosion intensity are not so extensive. According to Chiu et al. [38], in cavitation erosion, surface roughness (Ra parameter) increases up to limit value that is related to the steel grade with increasing testing time and then remains unchanged. However, Dojcinovic et al. [39] showed that the mean surface roughness Ra of ductile cast iron exposed to cavitation increases with testing time, although the rate of roughness change decreases. In the presented results, in cavitation erosion, surface roughness depends also on the cavitation intensity, i.e., flow velocity (Figure 12 and Figure 13). Taking into account the maximum Ra parameter obtained on the exposed surface after 600 min of testing and flow velocity, the correlations shown in Figure 21 a are obtained. As shown in Figure 21a, the Ra parameter increases with increasing flow velocity from 35 m/s to 38 m/s, but for flow velocities greater than 38 m/s, a decrease in surface roughness with increasing flow velocity was obtained. This decrease was probably due to the strain hardening (Figure 10b), the intensive development of surface damage and an increase in overlapping damage (Figure 16 and Figure 17). Regardless of the flow rate, higher roughness values were obtained for thermally treated steel, probably due to its lower hardness. Taking into account the strain hardening, the greater the increase in hardness the lower Ra parameter is observed only for cavitation erosion tests conducted at 43 m/s, but not at 35 m/s. Intense overlapping damage and intensive mass loss may cause a decrease in surface roughness. As a result, the surface roughness obtained at 43 m/s was lower than that obtained at 35 m/s.

In the case of slurry erosion tests, a linear increase in surface roughness with increasing rotational speed was obtained (Figure 21b). This correlation was independent of the steel hardness. A similar correlation was obtained by Abedini et al. [40]. As in the case of cavitation erosion, higher Ra values were obtained for thermally treaded steel that had lower hardness. Besides, the difference between Ra values increased with increasing rotational speed. Taking into account the changes in hardness (Figure 11b), the steel after thermal treatment underwent greater strain hardening than the steel in the as-received state. However, the surface roughness did not follow it. This was likely caused by more intensive mass loss (Figure 8). The lack of overlapping damage (Figure 18), as was the case with cavitation erosion (Figure 16 and Figure 17), contributed to a constant increase in roughness, not its decrease as in cavitation erosion.

## 5. Conclusions

The conducted investigations show that an increase in a flow velocity or a rotational speed increases the mass loss of the S235JR steel, but the nature of this increase is different for each type of erosion. In the case of cavitation erosion, this increase was a linear function of the flow velocity, and in the case of slurry erosion, it was a polynomial function. This is the first fundamental difference between cavitation erosion and sludge erosion, demonstrating the different nature of the destruction process.

The decrease in the hardness of S235JR steel due to the performed thermal treatment had a different impact on the process of cavitation erosion than on slurry erosion. In the case of cavitation erosion, the increased mass loss occurred immediately after the start of the erosion tests, regardless of the flow rate. In the case of slurry erosion, an effect of a decrease in hardness of the tested S235JR steel depended on the test duration and rotational speed. After the entire 600 min tests, it had slight effect on the mass loss, regardless of rotational speed. However, at the beginning of the tests, the influence of steel hardness on the mass loss depended on the rotational speed. With the increase of the rotational speed, an accelerated decrease in the mass loss of the softer steel in relation to the harder steel was observed. The rotational speed also influenced the duration of the test when the mass loss of the softer thermal-treated steel was lower than that of the harder as-received steel. Thus, the effect of steel hardness on mass loss is the second difference between these types of erosion.

With the increase of the erosion intensity, the strain hardening increased. However, regardless of the erosion intensity, the strain hardening was greater and deeper in cavitation erosion. At the lowest erosion intensity (the flow velocity of 35 m/s or rotational speed of 565 rpm), no strain hardening occurred in S235JR-AR steel, in opposition to S235JR-TT steel. At the highest cavitation erosion intensity (flow velocity of 43 m/s), the strain hardening increased to about 8% for the S235JR-TT steel and 10% for the S235JR-AR steel, and this maximum value was obtained at 235 µm below the surface. The entire depth of strain hardening was deeper for S235JR-AR steel (4 mm) than for S235JR-TT steel (2.5 mm). In the case of slurry erosion at a rotational speed of 1012 rpm, the strain hardening increased to about 7% for the S235JR-TT steel and 5% for the S235JR-AR steel, and this maximum value was reached 35 µm below the surface. The depth of strain hardening was 1.8 mm and 1.4 mm, respectively for the as-received and the thermally treated steel. This shows that the effect of erosion intensity on strain hardening is the third difference between these types of erosion.

The development of surface roughness (Ra parameter) was related to the erosion process and the intensity of erosion. In the case of cavitation erosion, the Ra parameter reached the maximum value in the tests conducted at 37.9 m/s, while a further increase in the flow velocity to 40.5 m/s and 43 m/s resulted in a reduction of surface roughness due to overlapping damage. In the case of slurry erosion, the greatest increase in surface roughness occurred in the initial stage of degradation. As the rotational speed increased, the roughness of the surfaces increased due to the lack of overlapping damage. Thus, the nature of the development of surface damage (presence or absence of overlapping damage) and changes in surface roughness are other differences between the types of erosion studied.

The conducted research indicates the need to exercise great caution when formulating theses regarding the influence of individual material properties on erosion resistance, as well as when over-generalizing the obtained dependencies. The results obtained in cavitation erosion may not be transferred to slurry erosion in any aspect of material degradation and vice versa.

## Figures and Tables

**Figure 1 materials-14-01456-f001:**
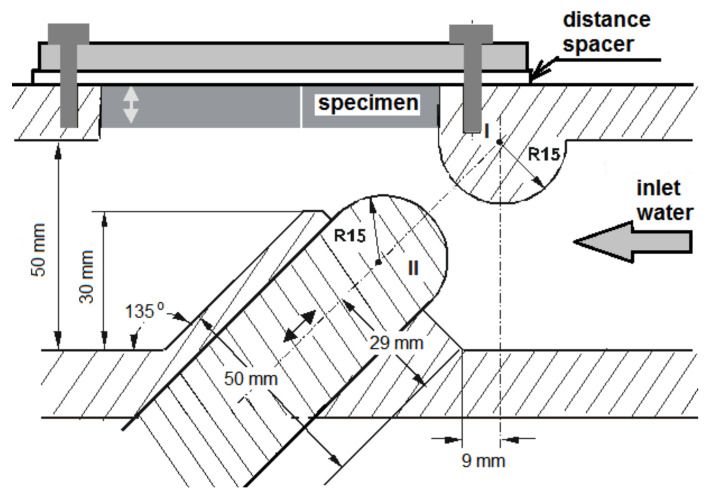
Scheme of the cavitation tunnel.

**Figure 2 materials-14-01456-f002:**
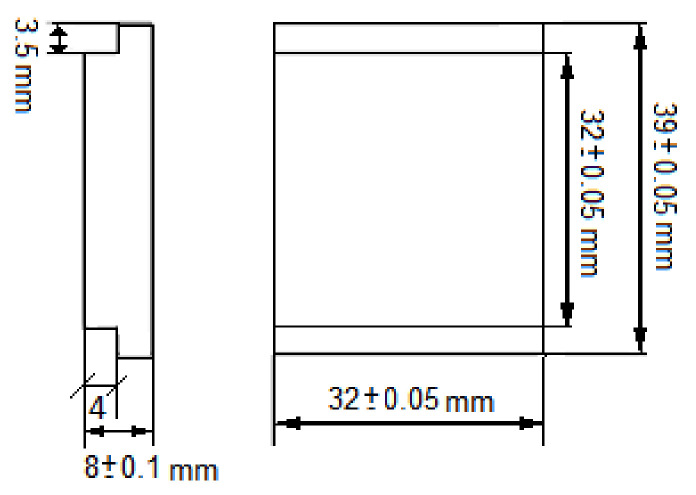
The shape and dimensions of a specimen for cavitation erosion tests.

**Figure 3 materials-14-01456-f003:**
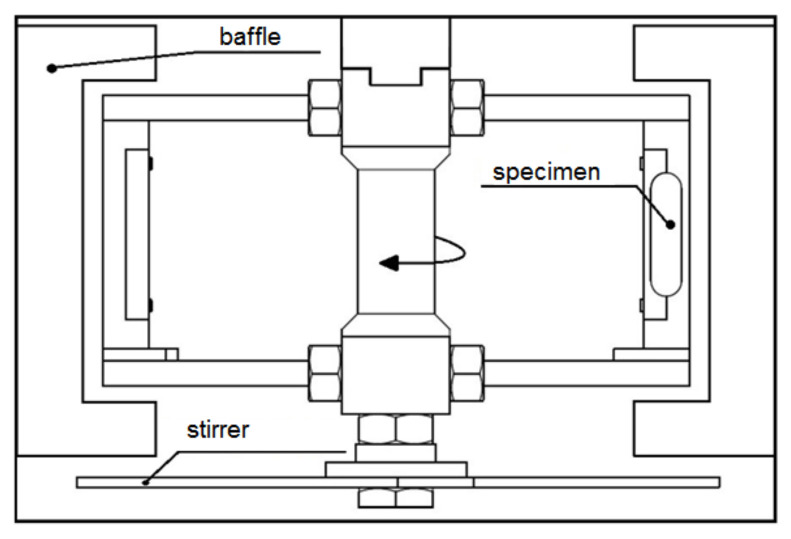
Scheme of a slurry pot tester.

**Figure 4 materials-14-01456-f004:**
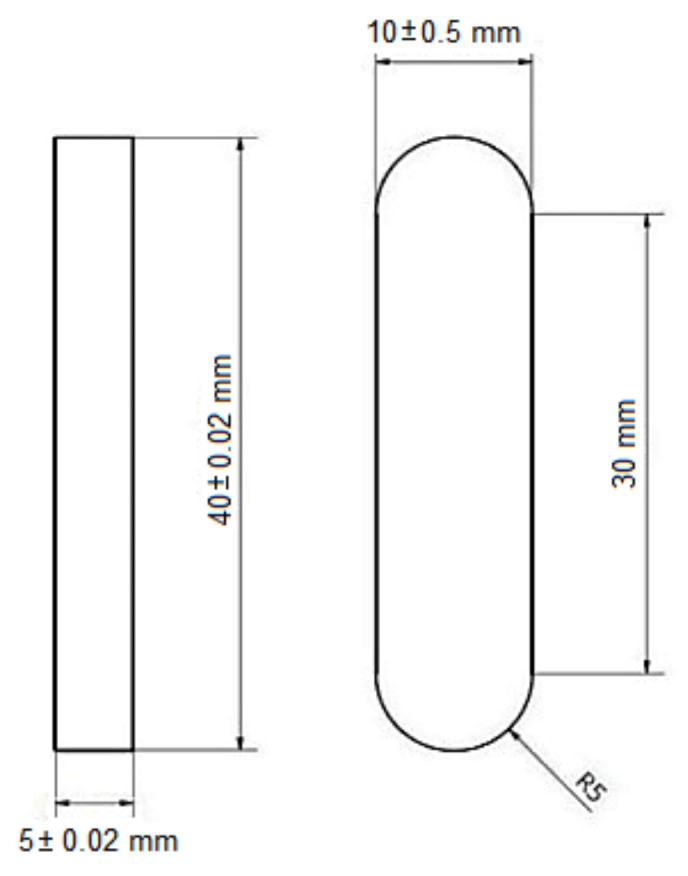
The shape and dimensions of a specimen for slurry erosion tests.

**Figure 5 materials-14-01456-f005:**
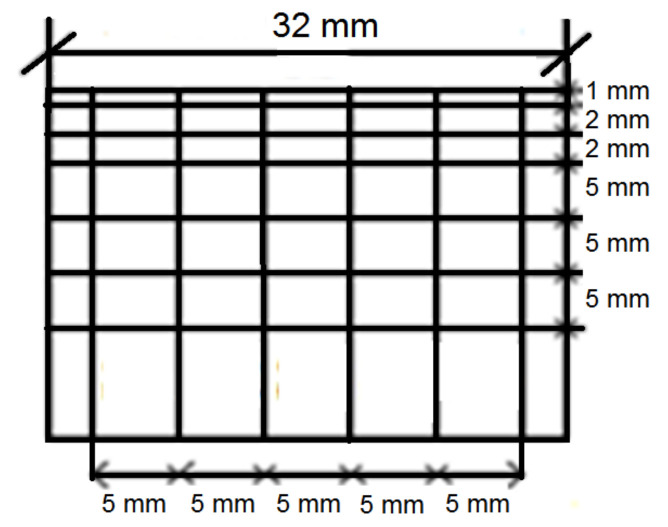
The scheme of the network for surface roughness measurements.

**Figure 6 materials-14-01456-f006:**
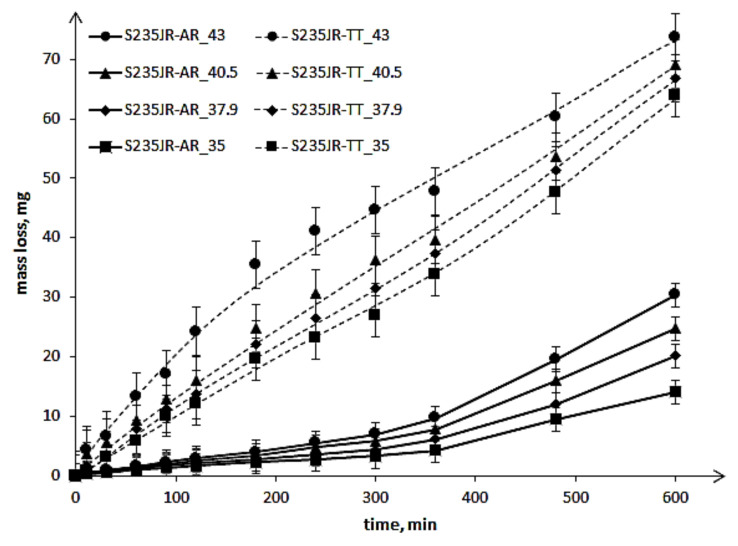
Cavitation erosion curves of S235JR steel in as-received state and after thermal treatment.

**Figure 7 materials-14-01456-f007:**
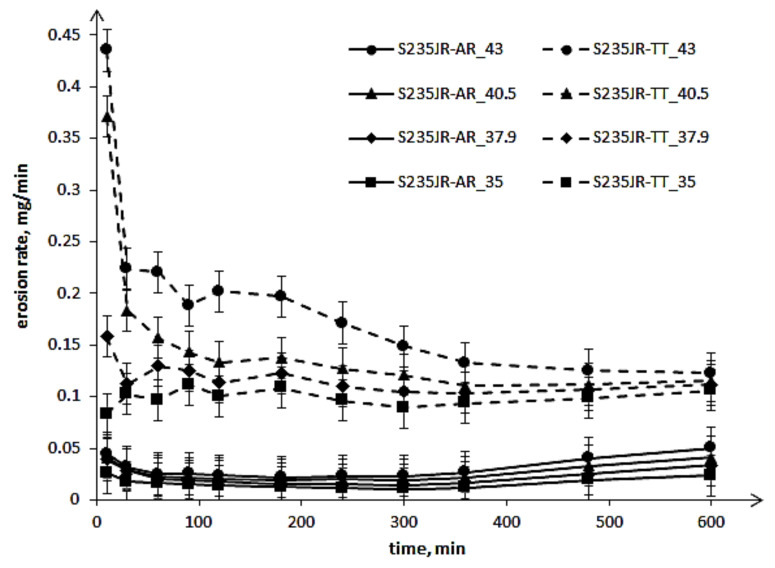
Cavitation erosion rate curves of S235JR steel in as-received state and after thermal treatment.

**Figure 8 materials-14-01456-f008:**
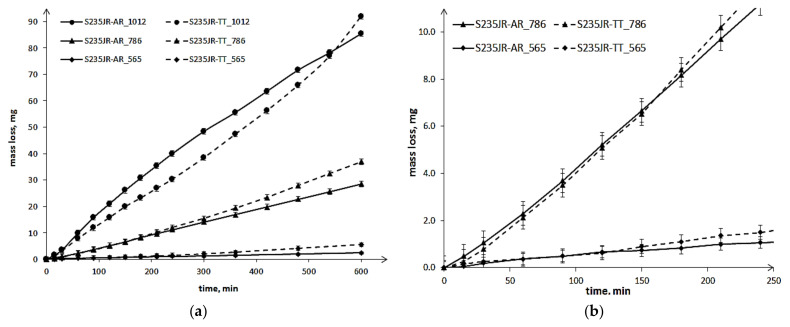
Slurry erosion curves of S235JR-AR and S235JR-TT steels: (**a**) full test duration, (**b**) initial 250 min of testing steels at 786 and 565 rpm.

**Figure 9 materials-14-01456-f009:**
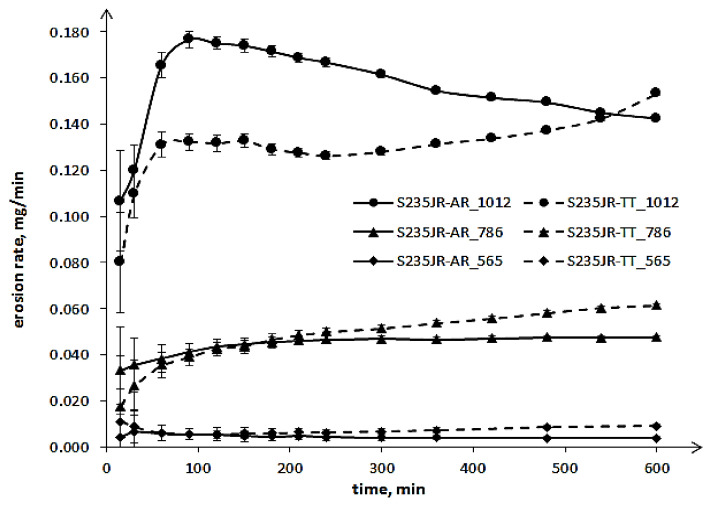
Cavitation erosion rate curves of S235JR steel in as-received state and after thermal treatment.

**Figure 10 materials-14-01456-f010:**
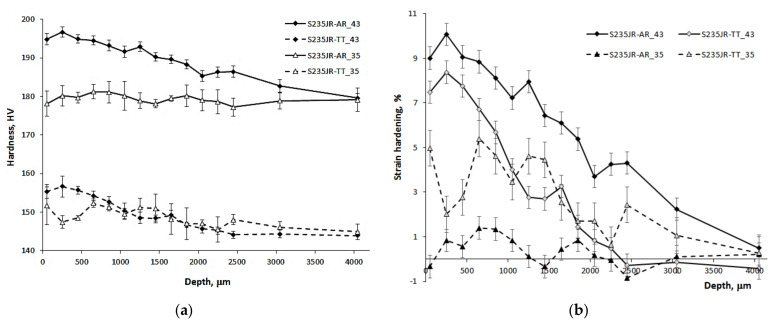
Hardness (**a**) and strain hardening effect (**b**) along the specimen depth caused by cavitation.

**Figure 11 materials-14-01456-f011:**
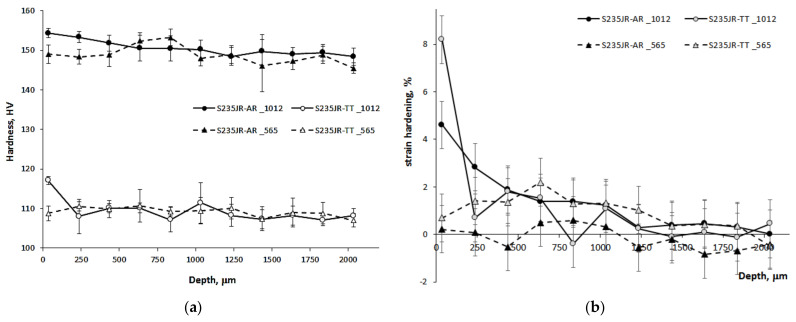
Hardness (**a**) and strain hardening effect (**b**) along the specimen depth caused by slurry.

**Figure 12 materials-14-01456-f012:**
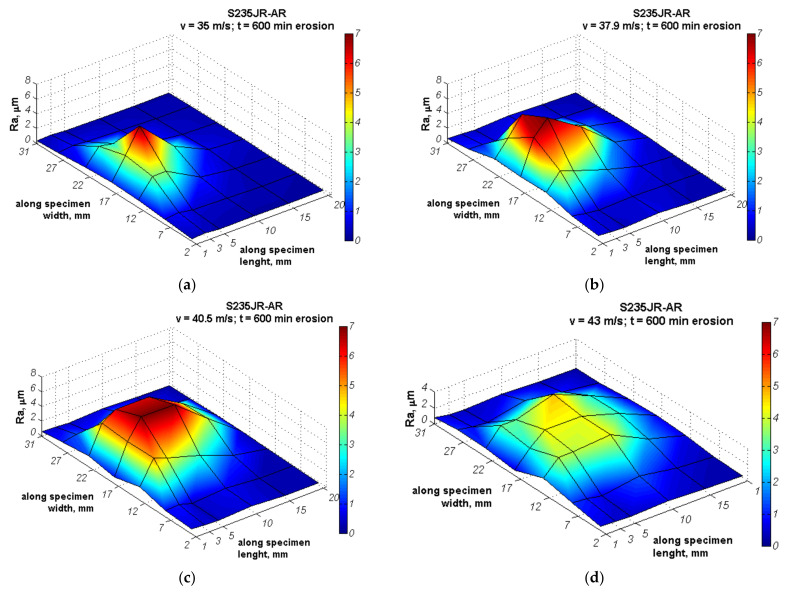
Distribution of the Ra parameter on the surface of the S235JR-AR steel tested for resistance to cavitation erosion at a flow velocity of (**a**) 35 m/s, (**b**) 37.9 m/s, (**c**) 40.5 m/s and (**d**) 43 m/s.

**Figure 13 materials-14-01456-f013:**
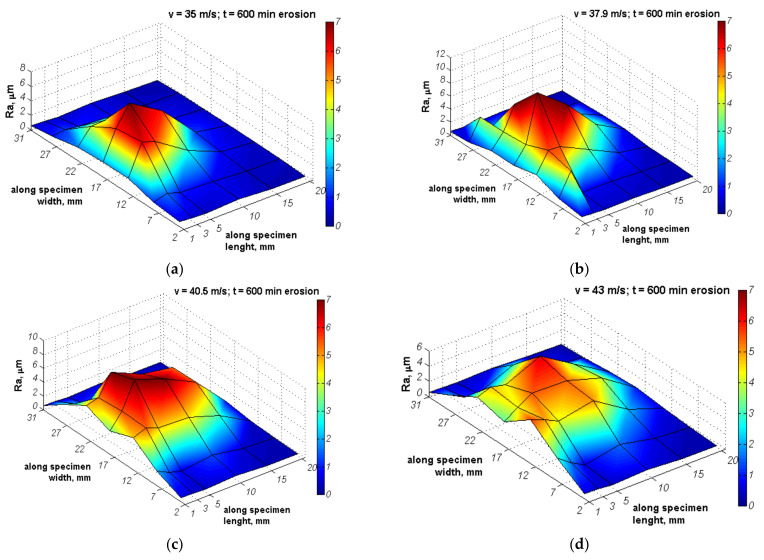
Distribution of the Ra parameter on the surface of the S235JR-TT steel tested for resistance to cavitation erosion at a flow velocity of (**a**) 35 m/s, (**b**) 37.9 m/s, (**c**) 40.5 m/s and (**d**) 43 m/s.

**Figure 14 materials-14-01456-f014:**
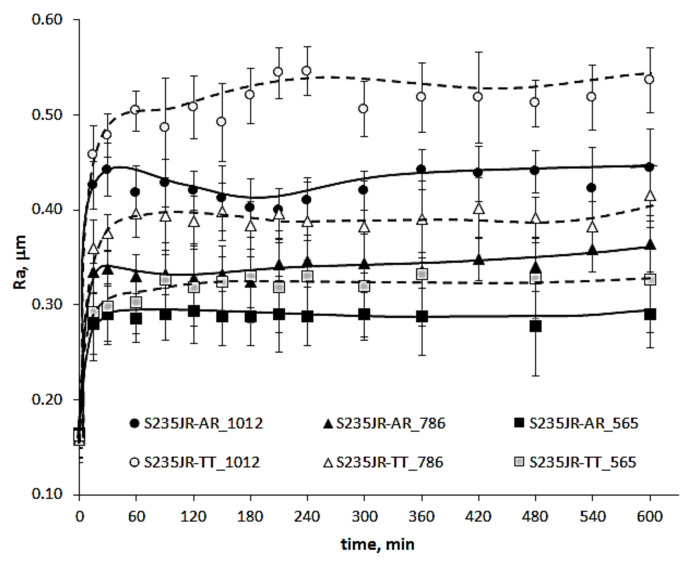
Development of surface roughness (Ra parameter) during slurry tests.

**Figure 15 materials-14-01456-f015:**
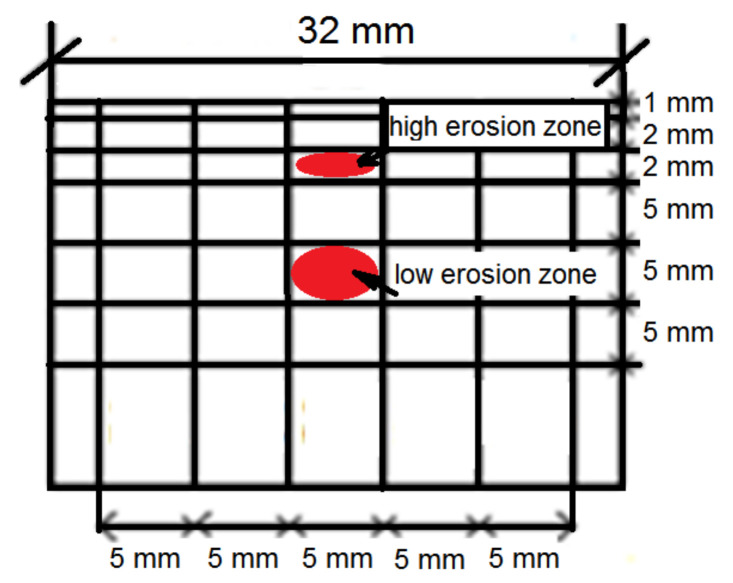
The observation areas of damage developed on the surface of S235JR steel during cavitation erosion tests.

**Figure 16 materials-14-01456-f016:**
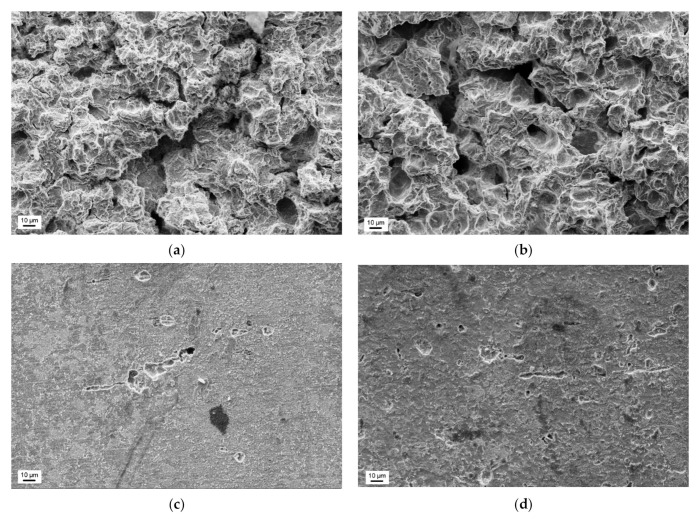
Surface damages developed during cavitation erosion tests carried out at a flow velocity of 35 m/s in the high erosion zone of (**a**) S235JR-AR steel and (**b**) S235JR-TT steel, and in the low erosion zone of (**c**) S235JR-AR steel and (**d**) S235JR-TT steel.

**Figure 17 materials-14-01456-f017:**
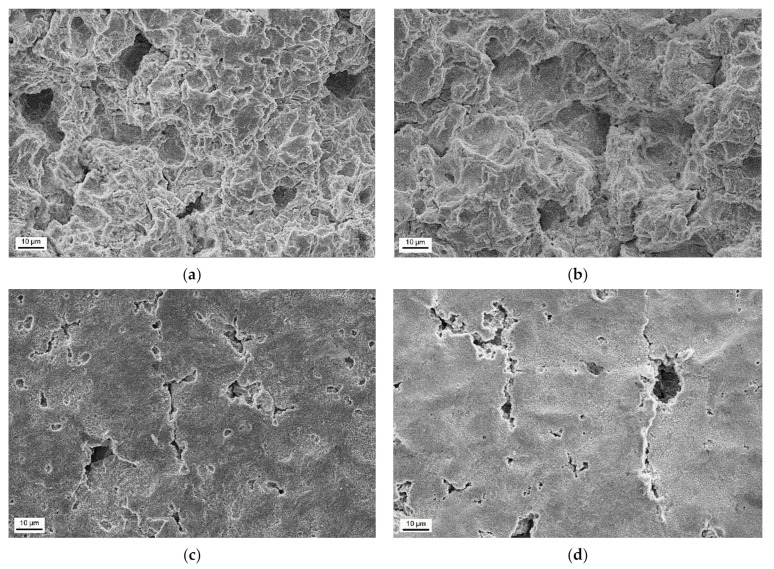
Surface damages developed during cavitation erosion tests carried out at a flow velocity of 43 m/s in the high erosion zone of (**a**) S235JR-AR steel and (**b**) S235JR-TT steel, and in the low erosion zone of (**c**) S235JR-AR steel and (**d**) S235JR-TT steel.

**Figure 18 materials-14-01456-f018:**
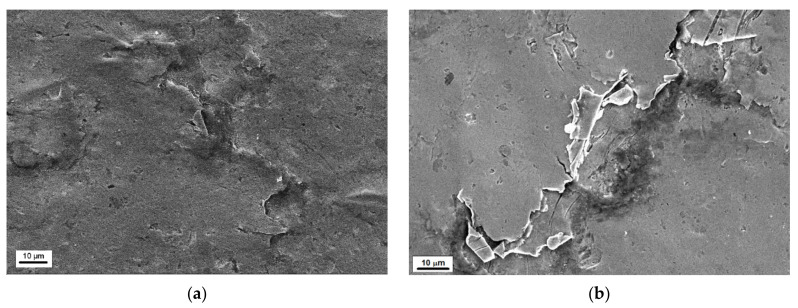
Surface damages developed during slurry erosion tests carried out at a rotational speed of 1012 rpm in (**a**) S235JR-AR steel and (**b**) S235JR-TT steel, and at a rotational speed of 565 rpm in (**c**) S235JR-AR steel and (**d**) S235JR-TT steel.

**Figure 19 materials-14-01456-f019:**
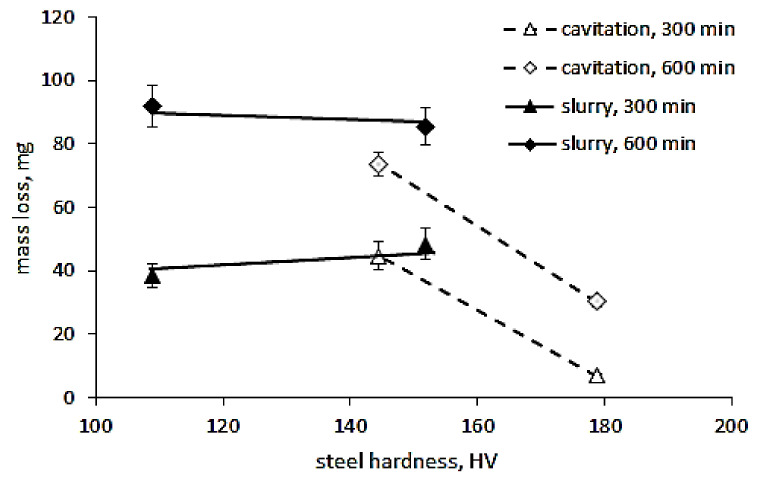
The effect of steel hardness on mass loss.

**Figure 20 materials-14-01456-f020:**
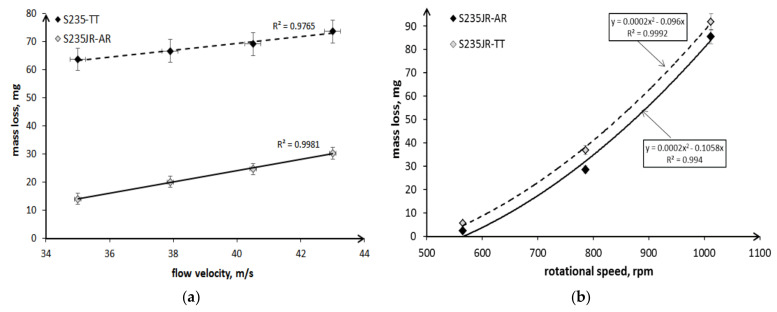
Correlation between the mass loss and (**a**) a flow velocity in cavitation erosion tests, and (**b**) a rotational speed in slurry erosion tests.

**Figure 21 materials-14-01456-f021:**
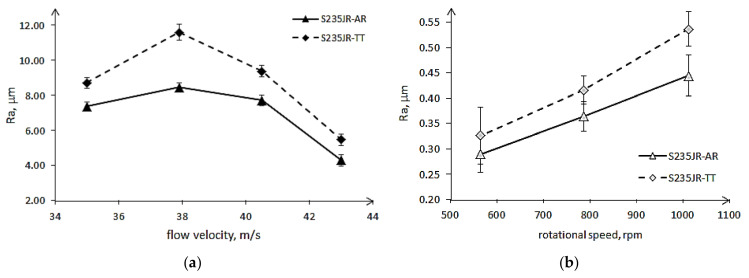
Correlation between (**a**) the maximum Ra parameter and a flow velocity in cavitation erosion tests, and (**b**) the mean Ra parameter and a rotational speed in slurry erosion tests.

**Table 1 materials-14-01456-t001:** Chemical composition of S235JR steel, wt.%, steel standard.

C	Mn	Cu	N	Si	P	S	Fe
<0.17	<1.4	<0.55	<0.012	<0.03	<0.035	<0.035	Rest

## Data Availability

Data available upon request due to privacy or ethical restrictions. The data presented in this study is available from the respective author upon request. Data is not publicly available due to additional research work being carried out.

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
