# Peer review of "Effect of Thermal Treatment and Erosion Aggressiveness on Resistance of S235JR Steel to Cavitation and Slurry"

_materials, 2021, doi:10.3390/ma14061456_

Round 1

Reviewer 1 Report

The main criterion why I tend to average is that it is not clear why exactly the investigated material and its heat treatment status was selected. Please clarify and sharpen what exactly is the scientific challenge of the work performed. At the beginning hardness seems of interest.
The tests and the results are presented clearly and understandably.
Since the initial motivation is unclear, the conclusions are not really meaningful and are more a summary of the results. The statement made at the end, that the two test methods are not transferable, is state of the art.

Author Response

Thank you for your comment. The motivation for selecting the investigated material and heat treatment has been improved, as well as the aim of this paper and the description of the results obtained. The scientific challenge of the work performed is the search for a correlation in the degradation of carbon steel caused by cavitation and slurry erosion based on the analysis of changes in mass loss/erosion rate in relation to changes in hardness (strain hardening), roughness and surface damage.

Reviewer 2 Report

First of all I would like to thank you for considering our journal to publish your manuscript. I have read your manuscript entitled Effect of thermal treatment and erosion aggressiveness on resistance of S235JR steel to cavitation and slurry with great interest. The paper contains material which is worthy of publication and of potential interest to readers, including the investigation of the impact of the change in the rate of cavitation and slurry erosion of carbon steel. The tests were performed at various intensities of erosive interactions with a high focus on the hardness which plays a key role in resistance to cavitation and slurry erosion.

The paper is very clear and well written, therefore the manuscript needs only some minor revision before it can be published. The points to be improved are mentioned below:

  1. In reference section, please use abbreviated journal name.
  2. In page 2 line 48, remove in Ref.
  3. In page 3 line 79, modify  has been described in Ref [17] with has been described by Krella et al. [17]
  4. In page 18 line 503, please change A similar correlation was obtained in Ref. [39] with A similar correlation was presented by Abedini et al. [39]
  5. Also in Discussions section in Refs is presented 4 times. Please remove/ change as described before.

Author Response

Thank you for your comments. All the mentioned points have been corrected.

  1. In reference section, please use abbreviated journal name.
    The abbreviations of journal name have been used.
  2. In page 2 line 48, remove in Ref.
    It has been removed..
  3. In page 3 line 79, modify  has been described in Ref [17] with has been described by Krella et al. [17]
    It has been modified.
  4. In page 18 line 503, please change A similar correlation was obtained in Ref. [39] with A similar correlation was presented by Abedini et al. [39]
    It has been corrected.
  5. Also in Discussions section in Refs is presented 4 times. Please remove/ change as described before.
    It has been corrected.

Reviewer 3 Report

The mechanism of cavitation and slurry is not investigated clearly. The manuscript is not of importance, science and originality.

Author Response

I'm sorry you have that opinion on this work and pointed out that neither the research was designed appropriate nor the results were clearly presented, but I cannot agree with it. The article presents very detailed studies showing the influence of heat treatment and the intensity of dynamic loads on the degradation processes of carbon steel. Due to the large amount of research presented, the description was perhaps too modest and hence your reception of this work. The article shows that the results obtained in one erosion process are not representative of the other in any aspect of degradation. Each analysed criterion of degradation of this carbon steel in both processes is completely different: different hardness causes a completely different effect in terms of the mass losses/erosion rates, strain hardening, surface roughness or development of surface damage. Of course, you can find papers on cavitation erosion and slurry erosion, but these are often papers on different materials. Besides, there are only presented one or two aspects of failure. At the moment, there are no articles covering all aspects of the degradation that have been analysed in this article. In this paper, the differences in the destruction processes are presented in-depth. This is an important aspect of the presented research, as it is still being said that the result can be interchangeable. Moreover, the obtained results of the steel hardness in slurry tests are opposite to those presented by Kahraman et al. [6], where the same steel was tested.